# Medium Extracellular Vesicles—A Qualitative and Quantitative Biomarker of Prostate Cancer

**DOI:** 10.3390/biomedicines10112856

**Published:** 2022-11-08

**Authors:** Milena Świtońska, Oliwia A. Jarosz, Dagmara Szołna-Klufczyńska, Katarzyna Sierakowska

**Affiliations:** 1Department of Neurosurgery and Neurology, Nicolaus Copernicus University in Toruń, Ludwik Rydygier Collegium Medicum, 85-168 Bydgoszcz, Poland; 2Faculty of Healths Sciences, Ludwik Rydygier Collegium Medicum, Nicolaus Copernicus University in Torun, 85-067 Bydgoszcz, Poland; 3Department of Anaesthesiology and Intensive Care, Department of Anesthesiology and Intensive Care for Children, Antoni Jurasz University Hospital No. 1, Collegium Medicum Nicolaus Copernicus University, 85-094 Bydgoszcz, Poland; 4Department of Anaesthesiology and Intensive Care, Antoni Jurasz University Hospital No. 1, Collegium Medicum Nicolaus Copernicus University, 85-094 Bydgoszcz, Poland

**Keywords:** medium extracellular vesicles, microparticles, prostate cancer, biomarkers

## Abstract

For years, the diagnosis of prostate cancer has been understated. Despite the relatively low mortality rate, prostate cancer is still one of the most common neoplasms in men, which proves the need for continuous improvements in the diagnostics of this disease. New biomarkers may address these challenges in the form of extracellular vesicles (EV) secreted by prostate cancer cells. The available literature in the PubMed, SCOPUS, and ResearchGate databases from the last ten years was analyzed using search phrases such as extracellular vesicles, microparticles, microvesicles, cancer biomarkers, and prostate cancer. Then, the research was selected in terms of the size of the tested EVs (the EV medium of 100–1000 nm diameter, was taken into account), the latest versions of the literature were selected and compiled, and their results were compared. The group of extracellular vesicles contain a substantial amount of genetic information that can be used in research on the specificity of prostate cancer and other cancers. So far, it has been shown that EVs produced by PCa cells express proteins specific for these cells, which, thanks to their specificity, can make EV useful biomarkers of prostate cancer. Moreover, the importance of the quantitative release of EV from PCa cells has been demonstrated, which may be necessary to diagnose prostate cancer malignancy. Each method positively correlates with Gleason’s results and is even characterized by greater diagnostic sensitivity. Medium extracellular vesicles are a promising research material, and their specificity and sensitivity may allow them to be used in future prostate cancer diagnostics as biomarkers.

## 1. Introduction

Prostate cancer is one of the most common malignant neoplasms in terms of morbidity among men and is at the forefront of mortality among cancer patients worldwide [1,2]. Cancer mainly affects patients over 50 years of age, and the probability of its occurrence increases with age [3]. According to statistical data, mortality as a result of prostate cancer is 10%, and the prognosis for early-stage detection is optimistic [4]. The standard diagnostic procedure in the case of prostate cancer includes imaging diagnostics, rectal examination, and determination of the PSA tumor marker in the serum [3]. While the first two procedures do not raise any objections, the determination of PSA is controversial due to the lack of strict specificity for prostate cancer, and its concentration may also increase with prostatic hypertrophy or inflammation; therefore, there is a risk of misdiagnosis [5]. 

In response to diagnostic problems of neoplastic diseases, there has been a significant amount of research on improving cell differentiation techniques in recent years. One of the most promising is the determination of potential prostate cancer biomarkers such as extracellular vesicles (EV). Extracellular vesicles appear among cancer cells almost immediately and fulfill the most critical functions enabling the growth, development, and expansion of the tumor. Moreover, both exosomes and ectosomes arising from the cells of a given tumor become a potential marker with a specific membrane phospholipid system, with characteristic proteins, as well as with valuable information inside, which is in the form of transferred genetic information [6,7,8]. The possibility of their quick detection and determination would indicate the presence of cancer cells, which would dramatically accelerate and clarify the diagnostic process of prostate cancer, thus allowing for prompt implementation of effective treatment. This, in turn, would increase the chances of a complete recovery. Studies have shown significantly increased levels of medium EV in many other diseases, including deep vein thrombosis, cardiovascular disease, diabetes, aplastic anemia, acute ischemic stroke, and infections [9]. However, these extracellular vesicles were produced by the body’s stem cells, either indicating a defect or containing information for the immune system. In the case of cancer, the production also occurs in all newly formed cancer cells. The level of presence and detection of EV in diseases brings promising prospects for new diagnostic models in a minimally invasive form, such as from the patient’s plasma. Innovative diagnostic methods are essential in cardiovascular, neurological, and oncological diseases [9,10].

The literature has been analyzed in PubMed, SCOPUS, and ResearchGate databases, including articles from the last ten years, using search terms such as microparticles, extracellular vesicles, cancer biomarkers, and prostate cancer. The latest versions of the literature were selected and compiled, and then studies that met the size criteria of the average extracellular vesicles tested (i.e., 200–1000 nm) were selected. Publications with a defective research methodology and lack of a precise description of both the selected particles themselves and the methods of their isolation and detection were rejected. The remaining articles meeting the requirements of the review were analyzed and compared.

## 2. Characteristics of Medium Extracellular Vesicles

Research on extracellular vesicles (EV) dates back to the mid-twentieth century. It was then suspected that human serum contained a prothrombotic subcellular procoagulant factor. A series of discoveries in this direction was initiated by Chargaff and West in 1946, distinguishing a “precipitating factor” in the platelet-free plasma, which was involved in blood coagulation processes [11]. Twenty years later, this accomplishment allowed for the first electron microscopy studies in 1967 by Wolf, in which he identified “platelet dust” [12]. The seemingly insignificant “platelet dust” turned out to be extracellular vesicles taking part in practically all processes at the cellular level. The involvement of EV in the disease was first demonstrated in idiopathic thrombocytopenic purpura [13], and a year later the presence of EV in the plasma of patients with advanced Hodgkin’s disease was documented, which was a breakthrough achievement in the work of clinicians on extracellular vesicles [14]. In 1980, research began on the presence and role of EV also being produced by healthy cells of the human body—including erythrocytes [15], monocytes [16], or human umbilical vascular endothelial cells [17]. However, it was not until 2005 that the Subcommittee of the International Society of Thrombosis and Hemostasis in Vascular Biology developed and published an official definition of a microparticle that allows for the precise definition of extracellular vesicles falling in the size range 100–1000 nm [18] (Figure 1).

In biochemistry, a significant number of names for extracellular vesicles can be identified depending on the size, the name of the composing cell, or differences on both the molecular and morphological levels. These nomenclatures are used interchangeably and may directly mislead or confuse the reader. In the literature, one of the most commonly used names for vesicles below 1000 nm is exosome. Also, vesicles with a diameter of 100–1000 nm are commonly referred to by researchers as microparticles, interchangeably with microvesicles. The nomenclature turmoil makes it difficult to reach the precise results of research on EV and confuses researchers [6,7,19,20]. Despite the theoretical definition of extracellular vesicle subtypes by size, exosomes (<100 nm), ectosomes (100–1000 nm), and apoptotic bodies (>1000 nm), it is currently not possible to set a hard-practical size limit in EV subtypes. The weak size boundary between exosomes and ectosomes (between 100 and 200 nm) does not allow for a clear definition of the size of the vesicle data. Moreover, current separation methods are not refined enough to extract particles down to tenths of a nanometer, which results in a lack of accuracy in isolating vesicles of a given order of magnitude. Therefore, under the recommendations of the International Society of Extracellular Vesicles, in the following article, EV with the magnitude of 100–1000 nm will be referred to as medium extracellular vesicles [19,20,21]. Medium size EV, as well as other extracellular products, were initially considered an unnecessary waste of cellular processes, but with time their great importance in transport and information transfer was discovered. EV are produced by both healthy body cells and those in which the pathological process takes place. Extracellular vesicles transport such biological elements as genetic material (both in the form of DNA and mRNA) or peptide chains specific to their parent cells. The lipids that make up the EV extracellular membrane are also characteristic, as they are analogous to stem cell membranes. The functions of the EV are as varied as the cells that produce them are diverse. They participate in coagulation, inflammatory and growth processes, as well as in angiogenesis and cellular neoplasm [6,7,10,22].

Over time, it has been proven that neoplastic cells release more extracellular vesicles than the rest of the cells [6,23]. Considering the fact that each cell releases its cellular material into the extracellular space, it automatically becomes a producer of specific biomarkers carrying specific genetic information. However, studies show that only some EV produced by prostate neoplastic cells are characterized by a unique protein sequence allowing for the declaration of their strict specificity for this disease, which makes it difficult to ultimately select them and use them for diagnostic purposes [6,24].

## 3. Formation of Medium Extracellular Vesicles

Medium EV (commonly called microvesicles or microparticles) are submicron particles secreted by plasma membranes in response to an internal or external stimulus, or through cellular self-regulation. Processes such as activation, damage, or apoptosis of the cell most often accelerate or additionally stimulate the secretion process in extracellular vesicles [8,25]. EV come from cells that include platelets, monocytes, endothelial cells, red blood cells, and granulocytes which have a broad spectrum of biological activity and have a significant role in many cellular processes including intercellular communication, immune mechanisms, apoptosis, and homeostasis [26,27]. So far, it has been documented that the release of medium-size EV into the extracellular space occurs through direct budding of the plasma membrane, followed by membrane cleavage and release of the newly formed particle into the cell matrix [6,8,20,25,28]. Research on the precise understanding of EV secretion shows that the formation of medium size EV (also known as microparticles) differs from the release of small EV (exosomes), and the presence of calcium ions plays an extremely important role in this process [8,20,21,25,26].

Not all mechanisms of EV formation in vivo are fully understood. Current knowledge of medium EV generation comes mainly from experiments on isolated or cultured endothelial cells [28,29]. In vitro, medium EV can be released from cultured cells through the action of a wide variety of cytokines and apoptotic stimuli. The cell membrane is made up of characteristic phospholipids. Its outer layer is made up of phosphatidylcholine and sphingomyelin, while the inner layer is made up of phosphatidylethanolamine and phosphatidylserine. Cell activation or apoptosis is associated with the rapid release of intracellular calcium ions by the endoplasmic reticulum. This increase in calcium causes a change in the transmembrane steady state. Phosphatidylserine is displaced outward, and cytosolic enzymes (i.e., calpain) are activated, leading to the cleavage of the cytoskeleton, simultaneously causing vesicle budding and release of medium EV from membranes into the extracellular fluid [26,27,30]. Cultured endothelial cells release medium EV as a result of deliberate production with the participation of ATP and single-stranded genetic material, so it can be concluded that the release of microparticles is tightly regulated [8,20,27].

## 4. Diagnostic Value and Methods for the Determination of Extracellular Vesicles

Exposure of the phospholipids and specific receptors on the EV surface causes them to also act as bio transmitters of inflammation, thrombosis, and angiogenesis [29]. EV are considered regulators of intercellular interactions and may play a significant role in the homeostasis and pathogenesis of many diseases. They participate in vascular function and inflammation by modulating the production of nitric oxide prostacyclin, stimulating cytokine release, and inducing tissue factor gene expression in endothelial cells, monocyte chemotaxis, and adherence to the endothelium [27]. EV may contain transcription factors, mRNA, and microRNA, which also suggests their regulatory role. Thus, EV are not only a miniature version of the cell membrane they come from but also a specific biological element due to their particular properties. In the plasma of healthy people, there are extracellular vesicles of various origins, and their composition may vary depending on the cell from which they originate and the type of stimulus that caused their formation [6,29]. With various pathologies, their number increases significantly. By measuring phospholipids such as phosphatidylserine and characteristic proteins contained in EV (antigens representative of their stem cells), it is possible to determine precisely which cells they came from. Medium EV can be identified in blood plasma by a solid phase uptake test (which can provide both quantitative and qualitative information) and flow photometric analysis. In flow photometry, EV are identified as fragments of 0.1–1 µm. (mainly through quantitative analysis) [29].

One of the most important advantages of diagnostics based on EV detection is that they are present in virtually all body fluids: blood and its derivatives, urine, cerebrospinal fluid, and even bronchoalveolar secretions [21,31,32]. As a result, collecting research material is minimally invasive or non-invasive compared to standard tumor diagnostic procedures such as biopsy or surgical removal of a tumor fragment. Moreover, thanks to the available sampling of the material, there is no risk of trapping cells from only one specific site of the tumor, and clinicians can monitor EV levels at regular intervals [10].

The basic form of isolation of particles with a diameter of 30 nm to 5 μm is differential centrifugation, which some researchers consider to be their gold standard [33]. In the case of extracellular vesicles, ultracentrifugation is the isolation method necessary for the precise selection of the material, which allows for particle fraction even below 30 nm. Unfortunately, the procedure itself is not precise enough, and the magnitude of the centrifugal force exposes the sample to contamination and decay of the tested particles, prompting researchers to constantly search for better alternative methods [33,34,35].

In attempts to isolate different-sized EV, the scientific world has turned to detecting, analyzing, and quantifying EV. The first and one of the oldest methods is flow cytometry, which in the nanoscale (flow nano-cytometry) is still one of the most established and popular methods of EV determination. It often serves as a comparative tool in research on other methods, e.g., nanoparticle tracking analysis (NTA) or dynamic light scattering (DLS) [34,36].

According to studies, flow cytometry is one of the most traditional methods of EV determination [10,32,37,38]. Flow nano-cytometry allows for the differentiation of EV by size, but also differentiation with the use of fluorescently labeled monoclonal antibodies [10,37]. The most important advantage of this method is the ability to differentiate the microparticles in terms of quantity and quality. The main disadvantage is that particles <0.5 µm may be undetectable, which might be acceptable in the context of the interlaboratory coefficient of variation; however, this disadvantage most likely prevents FC from being recognized as the gold standard for EV detection [9,32,38].

Immediately after flow nano-cytometry, one of the popular diagnostic methods is the enzyme immunoassay ELISA, which allows one to sort fluorescently activated cells. This test focuses on detecting a specific protein and then staining it according to the concentration of peptides. The ELISA model has as many supporters as adversaries, therefore it has not become the gold standard method of determining EV [9,10,32,37,39].

With time, the analysis using the NTA method, the size range of detected and visualized particles of 10 to 2000 nm, has gained popularity. In NTA, the laser beam transmitted through the suspension illuminates the extracellular vesicles, and the scattered light is focused microscopically, allowing the image to be recorded with a video camera [33,34,36]. This method, like many others, is criticized mainly for the lack of standardization; however, the number of advantages of NTA prompts clinicians to move away from traditional methods of EV detection in favor of NTA [33,36,40]. In combination with flow nano-cytometry and electron microscopy, NTA can determine not only the size but also the concentration of vesicles in the suspension, the process is faster and the sample itself is not heavily contaminated [33,36]. Finally, the NTA results were found to be consistent with the flow nano-cytometry results, but only after fluorescent staining of the sample [34]. On the other hand, comparing NTA to DLS, the former one guarantees higher resolution in the tested sample, however the latter one has an advantage in terms of greater repeatability [41]. It cannot be ignored that NTA is a very promising procedure and in many respects, it surpasses standard methods of EV detection.

All EV detection and determination methods require expensive specialized equipment, and the process is usually time-consuming and error-prone. These major problems preclude researchers from introducing EV detection into daily diagnostic practice that requires precise, rapid, and accurate results. Therefore, research continues to find the gold standard for the determination of EV.

The most considerable problem at the moment is the separation of the EV isolation and marking process, which needs more specialized equipment, expensive technology, and ultimately more time for sample testing. Potentially, microfluidic platforms and electrochemical chips can be the answer. The technologies behind them can solve the problem of the two-stage process and allow for the simultaneous selection of particles and then their biochemical analysis. To date, clinicians are studying their effectiveness on small extracellular vesicles (exosomes) in cancer diagnosis (Table 1).

Devices, which in recent years have opened up completely new cancer diagnostic possibilities, combine the isolating part and the analytical part of potential biomarkers. Microfluidic technologies embedded in the chips allow EV separation with low sample consumption, high output, and easy integration [46,47]. Thanks to the platforms, particles can be separated by membrane filtration and matrix sorting or by immunocapture [46,47,48]. Researchers show that compared to the traditional “gold standard” ultracentrifugation, the recovery rate is 8.9 times higher in a much shorter time [46].

The biggest technical novelty that is taking a huge step toward the future is the possibility of modularizing and integrating the detection and analytical parts. EV determination methods also based on microflows include fluorescence, colorimetric, electrochemical, magnetic, and surface plasmon resonance technologies [46,48]. Although microfluidic methods are very optimistic in their possibilities due to fast analysis, high sensitivity, low reagent consumption, and easy transfer, like all biotechnology it faces its problems [46,47]. According to researchers, a single form of detection is insufficient for the time being, and for the dual mechanism to be fully correct, accompanying equipment such as pumps or detectors are needed, therefore clinicians have to wait for compact versions that can be used in everyday practice [47].

Electrochemical nanosensors are of great importance in the detection of EV, and their advantages include ease of use, high accuracy, and reliable repeatability [48]. After a selective reaction of the analyte, biosensory platforms send an electrical signal with the recognized element, which transmits information about the detection by transduction [42]. The field of electrochemical biosensors is currently developing using nanomaterials such as carbon nanomaterials, graphene, metals, and their oxides, as well as inorganic structures [42,49]. Due to high sensitivity, good specificity, and, above all, easy modular integration, electrochemical biosensors have great potential for participation in future separation and analytical platforms [42]. So far, electrochemical platforms have been tested on small EV. Neoplastic diagnostics are waiting for the research to be extended to include medium EV.

Electrochemical and microfluidic platforms are extremely useful for the detection and isolation of exosomes. They respond to problems and difficulties such as traditional forms of separation and analysis of extracellular vesicles. Does the question then arise whether these platforms can be equally effective for forms such as medium EV? It remains to look for new developments proving the universality of platforms for all subtypes of EV.

## 5. The Role of Extracellular Vesicles in Neoplastic Processes

With the growing interest of scientists in inconspicuous particles, EV became the center of information about cancer cells. So far, EVs multiple functions have been described, including a transport function for mRNA, miRNA, whole proteins, and active oncogenes. Moreover, they participate in intra- and extracellular communication, which is specific for cancer, and they play an essential role in antiapoptotic and spore processes, promoting adhesion and proliferation [6,28,31]. One of the newly discovered functions of extracellular vesicles is their involvement in drug resistance through communication between cells already resistant to the action of a given drug and sensitive cells [50]. It should be noted that the greater the knowledge about the EV, the greater the opportunities for the scientific world to use them and as the EV get more and more specific, the more carefully they should be looked at and the more practical research value they constitute.

The presence of extracellular vesicles in peripheral fluids was demonstrated in healthy and oncological patients. However, the processes taking place with the participation of neoplastic cells occur significantly faster. EV made from neoplastic cells retain the properties and morphotic similarity to the stem cell as those that arise from healthy cells. Therefore, they have specific antigens allowing for their identification in the environment. Thanks to the surface proteins on the phospholipid membrane of EV, tumor cells easily reach other tumor foci and actively participate in the expansion of cancer on previously healthy tissues. Research has shown that by supplying tumor cells with nutrients and oxygen, EV broadly supports the angiogenesis process, which is critical for cancer survival and growth. Moreover, with the participation of EV, there is a horizontal transfer of mRNA and tumor RNA, which is the basis for its development [8,10,51,52]. The production of extracellular vesicles is intended not only to support disease progression and metastasis but also to create an environment with a minimized immune response to the immune system [6,51].

Scrupulous research on extracellular vesicles demonstrates their extensive transport functions. So far, it has been shown that EV carry not only genetic material in the form of mRNA, miRNA, rRNA, or DNA but also functional transmembrane proteins, membrane protein enzymes, cytokines, chemokines, lipids, and their receptors [8,39]. Depending on the transported load, EV modulate their basic function. By transferring genetic material, oncogene transcripts can be multiplied and then translated into proteins in the recipient’s cells. In this way, neoplastic cells from docetaxel-sensitive cells are modified into drug-resistant cells. On the other hand, healthy cells receiving the transferred material from tumor EV may start producing the oncogene transcript, promoting the proliferation of tumor proteins and particles [21,39,52]. EV carrying transmembrane glycoproteins present on the surface of cancer cells may stimulate other cells to produce proteins that break down the basal membrane of healthy cells, and EV carrying TRPC5 induces resistance to anthracycline in other cancer cells [39]. By transporting the epidermal growth factor receptor (EGFR), the extracellular vesicles significantly promote the growth of neoplastic cells, which enhances the spread and growth of tumors [8,31,51,52]. While the transport role of EV seems obvious, the method of fast transport of information and components remains unclear. Researchers agree that the interactions between extracellular vesicles and their target cells may depend on a specific ligand-receptor recognition model, which supports the theory of the formation of membrane parts complementary to the recipient on the surface of neoplastic cells [31]. One thing is for sure, the shedding of EV carrying a transport charge from tumor cells serves primarily to manipulate tumors to make them as susceptible to tumor growth and invasion as possible.

Environmental manipulation by EV can also be seen in terms of the immunomodulation triggered by a developing tumor. In the course of the research, clinicians noted multiple methods of tumor immunoprotection against the host’s immune system. The first method is the elimination of T lymphocytes by FasL or TNF-dependent induction of an apoptosis-inducing ligand [8,21,39]. Another technique is to switch off the T cell response by inducing the proliferation of the regulator T [8,21]. Moreover, EV containing TGF-β ligand may impair natural killer (NK) activation [21,39]. Researchers also observed that the fusion of extracellular vesicles formed from cancer cells with monocytes inhibited the differentiation of monocytes into antigen-presenting cells. As a result of this reaction, however, the activation and functioning of cytolytic T lymphocytes was inhibited by initiating the release of immunosuppressive cytokines [8]. During the observation, it was also assumed that EV from healthy cells can be captured by neoplastic cells, creating a kind of camouflage of lipids and proteins specific to a healthy cell membrane [8]. While searching for information on the development of immunotolerance, it was noticed that the role of EV in modulating immunity could potentially depend on the stage of tumor progression [52].

In neoplastic disease, a particular analogy to the course of venous and arterial thrombotic disorders has been observed. In healthy people, under physiological conditions, extracellular vesicles take part in both procoagulation and anticoagulation processes, preserving, in a way, the hemostatic nature of the environment. All fluctuations in the quantity and quality of the EV involved in producing thrombin manifest in blood coagulation diseases. This applies to both increased bleeding and the formation of blood clots. It is closely related to the increased EV derived from tumor cells, in which tissue factor (TF) is overexpressed, resulting in blood coagulation among patients [8,10,52]. It is not uncommon for patients with cancer to develop thromboembolism. It is not only the specific nature of EV of neoplastic origin, but their functions also significantly affect the entire pathological process. Research shows that the number of extracellular vesicles substantially correlates with the invasiveness of cancer and may indicate the disease’s stage in the diagnostic process. The more EV derived from neoplastic tissue, the more aggressive it is and able to process angiogenesis efficiently [8,21,39]. 

As shown by the excellent literature on the subject, the role of EV in neoplastic processes is extensive, every biochemical process of neogenesis takes place with the participation of extracellular vesicles. Therefore, it is not surprising that scientists see in them another promising source of not only knowledge but also solutions to tormenting science questions.

## 6. Diagnostic Efficacy of Extracellular Vesicles in Studies on Prostate Cancer

Scientific research is still full of imprecise information regarding the differentiation of extracellular vesicle subtypes. This is probably due to the difficulty of precisely separating such minuscule particles without exposing the samples to massive losses. The vast majority of researchers undertake extensive research of 30–1000 nm particles (exosomes and ectosomes) regardless of the differences in the structure and formation of both subtypes. All these treatments are aimed at finding the most promising, specific, and easily accessible prostate cancer biomarker. Lorenc T. et al. made an excellent summary of the current state of knowledge on the usefulness of small extracellular vesicles (exosomes) in the diagnosis of prostate cancer. The research showed that the greatest biomarker potential is hidden in the transported RNA and proteins transcribed on the surface of exosomes, such as B7-H3 or HSP71. Moreover, a potential predictive marker may be the exosomal androgen AR-V7 receptors, which respond accurately and selectively to antiandrogens. This emphasizes the value of RNA and transported proteins and that they should be given the greatest attention [53]. As the authors point out, small extracellular vesicles turn out to be a promising direction in the search for prostate biomarkers. In the process of searching for answers about particles larger than 100 nm, but still smaller than apoptotic bodies, in the following review, the main focus is on the overarching thread of medium-sized EV. Studies involving groups of extracellular vesicles containing particles from 100 to 1000 nm are considered in this review (Table 2).

Bertoli G. et al. in the most recent studies demonstrated the importance of microRNAs in cancer diagnostics based on the biomarker function of medium extracellular vesicles. Clinicians noted the role of miR-153, which modulates mRNA expression and mediates the regulation of basically all oncogenic functions, e.g., proliferation, migration, and differentiation. Moreover, the most important influence on the growth of neoplastic cells is probably the regulation of KLF5 expression. Scientists have shown that miR-153 can be found in both small and medium extracellular vesicles in the neoplastic microenvironment. In a study summarized in 2022, small and medium EV fractions were isolated from the DU145 and PC3 cell lines, and then the degree of expression of miR-153 in each of them was analyzed. It has been shown that miR-153 is present mainly in the medium EV of the PC3 line, which originated from stage IV prostate adenocarcinoma. Researchers also showed that the presence of miR-153 is particularly expressed in cells with a high Gleason score. Furthermore, silencing the microRNA with oligonucleotides resulted in a reduction in the control of tumor cell proliferation. While looking for a suitable biomarker for prostate cancer, scientists also showed a path leading to the inhibition of tumor spread, which is an extremely promising direction [54].

In turn, the research carried out in 2016 showed diagnostic possibilities from the quantitative perspective of medium extracellular vesicles of cancer origin. Biggs C.N. et al. identified the plasma level of prostate medium EV in PCa patients and compared them with the level of PSA. Due to nanoscale flow cytometry’s effectiveness, they showed a high amount of PMP (prostate cancer microparticles) in patients with a Gleason score > 8, but they did not establish the same relationship in those with lower values of this score. It has been proved, however, that the plasma of patients with high-grade cancer contained significantly more medium extracellular vesicles than the plasma of remaining patients. According to the Gleason scale, the number of EV in the plasma correlated with the patient’s risk assessment. Medium EV levels were independent of PSA and decreased significantly after prostatectomy three weeks after surgery. A statistically higher level of EV has also been demonstrated in patients with refractory prostate cancer. Importantly, PSMA antibodies were used to identify medium extracellular vesicles in this study, the specificity of which for prostate cancer cells is questioned by the authors. This position is backed up by other studies showing that PSMA antibodies are expressed in cancer cells of different types, including kidney cancer, as well as squamous cell carcinoma of the oral cavity [55,56]. Moreover, studies show the production of PSMA antibodies also by healthy cells. This may explain the high number of PMP among patients with BPH (benign prostatic hyperplasia), which is also the most significant disadvantage of PSA diagnostics. The authors of the study themselves suggest the need to add another biomarker to refine the diagnosis of prostate cancer in the above-mentioned method. To sum up, the quantitative and qualitative determination of medium EV in patients with prostate cancer could serve as a population-based screening and prognostic way for early cancer recurrence after prostatectomy, and the above-mentioned study shows new possibilities in obtaining and using biomarkers for prostate cancer. However, it requires significant refinement and selection of diagnostic material [57].

**Table 2 biomedicines-10-02856-t002:** List of biomarkers in types of extracellular vesicles.

Authors	Type of Particle—A Biomarker of Prostate Cancer	Study Purpose	Specific Element	Research Method
Sandvig K. et al. [6]	Medium EV	PCa diagnostics by qualitative methods using EV	CDCP1 and CD151	Nanocapillary liquid chromatography
Bertoli G. et al. [53]	Medium EV	PCa diagnostics by qualitative methods using EV	miR-153	Medium EV Isolation by Differential Ultracentrifugation and miRNA extraction was performed by the Trizol method
Biggs. CN et al. [56]	Medium EV	PCa diagnostics by qualitative methods using EV	PSMA	Nanoscale flow cytometry
Tavoosidana G. et al. [58]	Medium EV	PCa diagnostics by qualitative methods using EV	five different epitopes on at least four different proteins	Expanded variant of the proximity ligation assay (PLA)—4PLA
Albino D. et al. [59]	EV (medium size with small size contamination)	Functions of vesicles in carcinogenesis and metastasis	miR-424	Isolation of EV, determination of the number of proteins from centrifuged plasma by BCA
Davey M. et al. [60]	EV (medium size with small size contamination)	Development of a biomarker panel based on EV detection	miR-375 and miR-574	Isolation of EV using centrifugation and the Vn96 Peptide
Brzozowski J.S. et al. [61]	EV (medium size with small size contamination)	Lipidomics analysis to quantify molecular species of EV lipids	sterile lipids, sphingolipids and glycerophospholipids	Nanoparticle Tracking Analysis (NTA)
Park Y.H. et al. [62]	EV (medium size with small size contamination)	PCa diagnostics by qualitative methods using EV	PSMA (+)	Transmission electron microscopy

The analysis by Sandvig K. and Llorente A. proves the existence of medium extracellular vesicles produced by PC-3 cells, which are EV involved in metabolic processes and intercellular transport. During the research, two proteins contained in medium extracellular vesicles showing extraordinary specificity were identified: CDCP1 and CD151, in the context of prostate cancer (Figure 2). It has been demonstrated that metastatic cancer cells have a higher content of CSCP1 protein compared to the cells of the non-metastatic cell line of both PC-3 and LNCaP. On the other hand, expression of the CD151 protein, in addition to the biomarker function, allows the prognosis of a low-grade neoplasm more effectively than the histological classification. The most significant advantage of this discovery is the specificity of the above-mentioned proteins contained in EV to prostate cancer. As research develops, they turn out to be more and more promising and bring greater benefits than assumed [6].

The evidence of the diagnostic significance of the CD151 protein contained in medium EV was documented by Ang J., Lijovic M., Ashman K. et al. These researchers showed that expression of CD151 protein positively correlated with the progression of prostate cancer and worse prognosis of patients. Moreover, it was proved that the predictive value of CD151 protein differentiation was significantly better than the traditional Gleason classification. This was confirmed by the research of the above-mentioned Sandvig and Llorente [6]. Researchers agree that there is a significant need for further research on the practical use of this diagnostic and prognostic form in prostate cancer [58].

Researchers led by Gholamrez Tavoosidan developed a modified proximity ligation assay (PLA) to detect medium extracellular vesicles in blood plasma of patients with prostate cancer. In the study, they compared a study group of patients with prostate cancer with a control group of people without cancer. The detection with four different mono- or polyclonal antibodies had a decisive advantage in the recognition of specific extracellular vesicles. In each vesicle, five different epitopes on at least four different proteins were recognized. This method proved to be highly sensitive and specific in a test that detected a significantly increased number of EV from cancer cells (which the researchers called prostasomes) in the blood of patients before radical prostatectomy, distinguishing them from patients with innocuous biopsy results. Moreover, the 4PLA test allowed to distinguish patients according to the advancement of the neoplastic process and showed that in patients with a Gleason score of 7 and 8/9, the median EV level was 2.5 to 7 times higher than those with a score of ≤6 according to this scale. In addition, researchers noted that EV was not detected by the 4PLA method in ultracentrifuged samples, unlike those that had not been subjected to the preliminary mechanical selection process. The whole study can be described as a promising prostate cancer diagnostic test based on biomarkers in the form of specific medium extracellular vesicles—prostasomes [59].

The most recent research into the use of potential prostate cancer biomarkers was carried out on extracellular vesicles containing miR-424. According to the authors of the 2021 study, this type of EV is easily detectable in plasma-type fluids and is specific for primary tumors and metastatic prostate epithelial cells. It has been documented that the released miR-424 via EV serves the neoplastic cells to transmit oncogenic signals both in the tumor microenvironment and in metastatic sites. EV contains a specific genetic material of the tumor. Albino D. et al. emphasize, however, that their research was mainly aimed at identifying the abovementioned EVs and characterization of their functions, and that the methods of EV detection containing miR-424 and research on their specificity and diagnostic utility are being left for further investigation on this subject while emphasizing the promising prognosis [60].

The overarching goal of researchers Davey M. et al. was to develop a panel of biomarkers based on the detection of EVs capable of distinguishing prostate cancer from mild inflammation. In total, 56 patients were enrolled in the study, including 28 people with prostate cancer (diagnosis based on tissue biopsy results) and 28 people without cancer. Researchers isolated particles 30 to 1000 nm in diameter from the urine and identified a panel of seven mRNA and miRNA biomarkers using the Vn96 peptide affinity method. The specific expression ratio showed a sensitivity of 75% and a specificity of 84%. Clinicians show that each miRNA turned out to be an effective biomarker with high predictive power for prostate cancer, but the greatest diagnostic capabilities of Vn96 were those of miR-375 and miR-574. Moreover, the researchers confirmed the fact that samples with Gleason scores above 7 are much easier and better captured from the entire pool. The usefulness of the miRNA contained in extracellular vesicles is emphasized by the fact that in the control group, there were patients with other inflammatory diseases of the prostate, in which the PSA test result could turn out to be a false positive. When using the quantitative polymerase chain reaction, there was no doubt as to the origin of the samples, which raises high expectations for both the method and the biomarkers themselves [61].

The fact that extracellular vesicles are produced by all cells of the body of healthy people as well as by cancer cells of sick people is obvious at the moment. Researchers focused their attention primarily on the specific genetic material contained in them, a potential cancer biomarker, but not only miRNAs contain valuable information allowing the detection of cancer-related EV. Brzozowski J. S. and his co-researchers show evidence that even EV lipids in healthy people from EV lipids in prostate cancer patients differ enough to be a good cancer biomarker. The team isolated EV from non-cancerous (RWPE1), cancerous (NB26), and metastatic (PC3) cell lines. The extracted lipids were subjected to quantification of molecular types of lipids. In the series of neoplastic origin, 187 molecular species of lipids different from those found in EV of non-neoplastic origin were recorded, and their composition was quantitatively dominated by sterile lipids, sphingolipids, and glycerophospholipids. Researchers emphasize the fact that lipid rafts may affect the selective sorting of proteins to EV, which distinguishes their importance in the structure of vesicles. The considerable role of lipids and the differences in their molecular structure forces us to delve into the subject further to find the ideal biomarker of prostate cancer [62].

Park Y.H. et al., in their extensive research, showed extracellular vesicles specific for prostate cancer that could be isolated from plasma collected by liquid biopsy. Researchers emphasize the methods of obtaining current biomarkers in EV, considering how difficult and complex material plasma is for selective diagnostics. Using aqueous two-phase methods, they successfully isolated PSMA-positive microparticles showing an extremely particular specificity for prostate cancer, allowing it to be differentiated from benign prostate hyperplasia with a sensitivity and specificity of about 91.7%. What is worth emphasizing, a correlation was also demonstrated between a low level of extracellular vesicles and a low Gleason score among patients, and the study showed a relationship between a low concentration of extracellular vesicles and better patient survival and absence of tumor recurrence [63].

## 7. Multiple Applications of Extracellular Vesicles

Many studies show that extracellular vesicles are promising candidates for prostate cancer biomarkers. Information about the vast, unused potential of the EV is repeated many times, which will result in discoveries in the following years. As one can see, marking EV as tumor biomarkers is only a fragment of their possible use [24].

Research demonstrated that using extracellular vesicles for drug transport can potentially increase drug cytotoxicity, as in the paclitaxel study. Taking a closer look at such materials as medium EV, other possibilities of using them were also noticed, for example, in new cancer treatment strategies [27,64,65]. This discovery could revolutionize the supply of chemotherapeutic agents, increase their direct effectiveness, and reduce the side effects of their use, but at the moment, there is too much risk in the design of clinical trials with their involvement. This is because the increase in the viability of neoplastic cells as a result of the supply of EV not saturated with the drug was shown. There is also the issue of protecting the drug-transporting particles from premature destruction by healthy phagocytic cells [64]. As other studies also show, the number of potential benefits of using EV to deliver therapeutics directly into the cancer cell would significantly change the meaning of the term targeted therapy and drastically reduce the side effects of traditionally administered chemotherapy. However, at the moment, much more extensive research on this therapeutic method is needed to eliminate all adverse effects, such as side activation of oxidative stress pathways, elimination of EV by the immune system, or the issue of 100% saturation of extracellular vesicles with the drug [65].

The abundance of possible uses of EV is unveiled with each new research in this area. Currently, scientists do not limit themselves only to the presence of EV, but also explore the possibilities of their use. As one can see, the most real biomarker function of extracellular vesicles is just the beginning of their participation in the fight against prostate cancer.

## 8. Discussion

The search for new cancer biomarkers has been going on for years. Researchers place their greatest hopes on miRNA, PSMA, and specific membrane proteins. All these elements are carried in the intercellular transport by extracellular vesicles. It can be noticed that the specificity of the tests is more often determined by specific elements contained in the vesicles and the tissues themselves than the actual EV [66]. However, it should be noted that a specific diagnostic value also lies in the quantitative production of vesicles and the observation of significantly increased EV secretion by cancer cells compared to other cells is considerable, and in the further course of the research, it may turn out to be of breakthrough value [6,23,57,67]. 

Besides the exclusive selection of EVs as cancer biomarkers, forms of obtaining them remain debatable. The question of bringing a new diagnostic source as quickly, efficiently, and cheaply as possible does not lose its importance, and the choice of methods—although of considerable size—does not facilitate the decision-making. It is difficult to find a diagnostic compromise when isolating PCa extracellular vesicles. Often, fast techniques are associated with many disadvantages, such as low purity of the obtained sample, laboriousness, or high costs. Acquisition of EV at this point focuses on three main techniques: differential centrifugation/ultracentrifugation, size exclusion chromatography (SEC), and ultrafiltration. The first of them, used most often, is carried out relatively easily, efficiently, and quickly. However, it is still characterized by the lack of scaling and low purity due to contamination of the sample with lipoproteins of similar size. On the other hand, the SEC method, second most frequently used option, is relatively fast, inexpensive to use, and with high purity and repeatability. However, like its predecessor, it is not scaled. Ultrafiltration is characterized by speed and low EV losses but lacks scaling and reference protocols [43,44]. All of the above methods undoubtedly need to be refined and validated on a commercial scale. 

The need to discover and implement new diagnostic tools is unquestionable; however, the role of the controversial PSA has not yet been decided. Scientists are looking for solutions to maximize the potential of a non-specific PSA and are testing new solutions to improve the current diagnostic process. Manipulation of the PSA factor in the prostate health index and the 4Kscore method mainly compares total PSA, free PSA, intact PSA, and other factors. In turn, transformations and modifications of the PSA factor itself take place in the glycosylation method, which supports the increase in the predictive value. Ways of using PSA in various forms bring back hope for clarification and then use the already known and relatively easy cost for the diagnostic option [66].

## 9. Conclusions

Medium extracellular vesicles have great potential in the future diagnostics of prostate cancer and other malignant neoplasms.

So far, higher diagnostic efficiency of medium extracellular vesicles has been demonstrated compared to the standard of PSA determination, which allows for improving the current diagnostic methods of prostate cancer.

Tests for the content of specific proteins in EV have an extremely high diagnostic potential. However, quantification of EV in prostate neoplasms also requires further prospective studies.

## Figures and Tables

**Figure 1 biomedicines-10-02856-f001:**
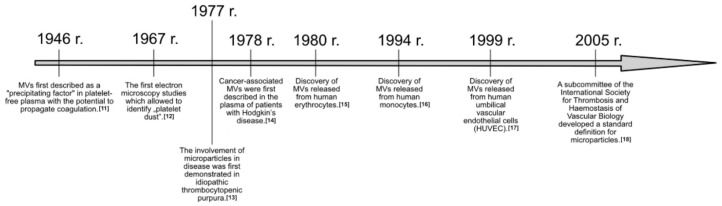
Timeline of the significant discoveries in extracellular vesicles research [11,12,13,14,15,16,17,18].

**Figure 2 biomedicines-10-02856-f002:**
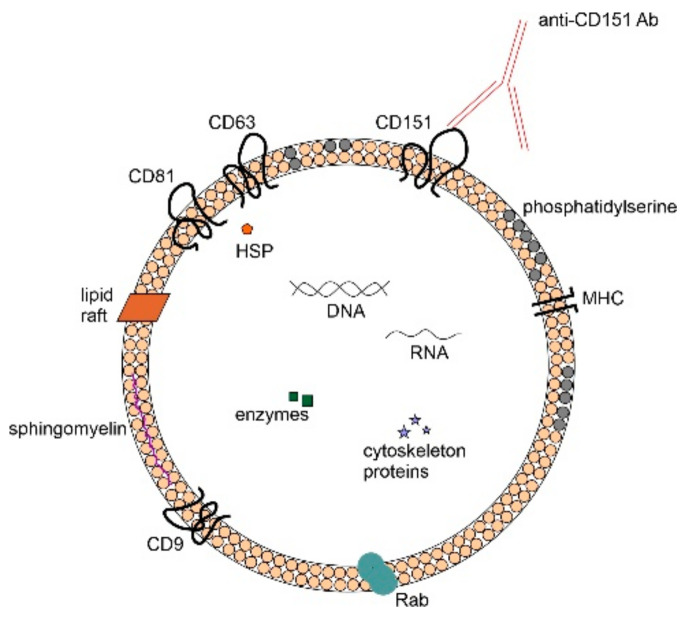
CD151 protein present in the membrane of the extracellular vesicle, which is a specific biomarker of prostate cancer.

**Table 1 biomedicines-10-02856-t001:** Key information about methods of detecting extracellular vesicles.

Detection Method	Type of Particle	Particle Detection Method	The Biggest Advantage	The Biggest Disadvantage	Bibliography
NTA	EV (all subtypes)	Visualization by light scattering with a light microscope. (individual visualization)	Analyzes the size distribution of monodisperse and polydisperse samples	No standardization	[35,39,40]
DLS	EV (all subtypes)	Visualization by light scattering with a light microscope. (Detection of intensity change)	High repeatability, simple and convenient to use	Inaccurate for polydisperse samples, the presence of large-size particles distorts the result	[35,40]
Microfluidic platforms	Small EV	Depending on the structure (Fluorescence, Colorimetric, Magnetic, Electron resonance vibrations, Electro-chemical, Immunological affinity)	High sensitivity, low consumption of reagents, very low sample contamination	Requires connection to some instruments such as syringe pumps and fluorescent detectors	[41]
Electrochemical platforms	Small EV	Depending on the construction of the biosensor (electro-chemical, photoelectro-chemical and colorimetric)	Combining exosome separation platforms with accurate exosome quantification technology	No information available on the selection of nanomaterials and how they interact with each other.	[42]
Flow nano-cytometry	EV (all subtypes)	Measurement of scattered light or fluorescence signals emitted by appropriately irradiated cells	Does not require isolation or concentration of EVs prior to staining	Conventional cytometer scan underestimates the amounts of particles smaller than 300 nm	[43]
ELISA (Enzyme-Linked Immunosorbent Assay)	EV (all subtypes)	The formation of bonds between the antigen and the antibody, which is revealed by a color reaction	Characterization and quantification of EVs isolated from a small amount of plasma	If the formulations are not well purified, co-isolated contaminating biomolecules could result in a high background levels of (biological noise)	[44]
Electron microscopy	EV (all subtypes)	It uses an electron beam for imaging	It has a resolution around 0.5 nm and it may provide detailed structural information of EVs	It cannot show EVs in their native state because the samples need to be fixed and processed prior to imaging	[45]

## Data Availability

Not applicable.

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
