# Peer review of "Medium Extracellular Vesicles—A Qualitative and Quantitative Biomarker of Prostate Cancer"

_biomedicines, 2022, doi:10.3390/biomedicines10112856_

Round 1

Reviewer 1 Report

The topic of review by Switonska et al. is useful. Overall content is informative and relevant however, it needs extensive revisions as per comments below:

General comments:

1. The choice of language is very poor. The authors should be careful with "synonym replacement". E.g. page 5 line 195 "oncological patients" is one such example. There are numerous such inappropriate synonym replacements. Authors should revise their write up extensively

2. More figures should be included. A figure summarizing the methods for exosome/microparticle/microvesicle, whichever term authors choose finally (see below my comment), detection should be included. Another "timeline figure" should summarize the significant discoveries chronologically.

Specific comments

1. The choice of term "microparticles" is ambiguous and confusing, slightly arbitrary. Authors should choose one term (e.g., micropartciles, or microvesicales), cite adequate references to support their choice and then stick to it throughout text

2. The paper is not formatted like a review paper. Also section numbering needs to be carefully checked

3. Page 3 last pargraph (line 129-142). The information is inaccurate. Not all exosomes need external stimulus for release. Extensive literature is available on this topic and many excellent reviews have been published. Authors should consult published papers (e.g., including but not limited to the excellent review by Hessvik et al. Cell Mol Life Sci. 2018; 75(2): 193–208)
4. Page 4 section 3.4: NTA is now much more commonly used. Authors should include more discussion on NTA based exosome detection. Also, a brief discussion on modern electrochemical and microfluidics based platforms will also be useful

5. Page 4 line 187: Traditional centrifugation is misleading term, plus this is not a detection method. Ultracentrifugation is needed to isolate exosomes. They are then detected by any of the several methods

6. Section 3.5 is incomplete. This section should highlight in more detail how exosomes/microparticles play role in neoplastic processes, e.g., the possible role played by exosomal cargo (this information is included in first paragraph of 3.6 but it belongs to 3.5)

7. There is a large body of research available on the role of exosomes in prostate cancer (e.g., an excellent recent review Int. J. Mol. Sci. 202021(6), 2118). Authors have only cited 7 studies which is not sufficient. This section should be expanded to inlcude all the significant contributions done so far in this area.

Author Response

Answer for the Reviewer

We would like to thank the Reviewer for very valuable comments. Detailed answers to individual items are provided below.

The topic of review by Switonska et al. is useful. Overall content is informative and relevant however, it needs extensive revisions as per comments below:

General comments:

  1. The choice of language is very poor. The authors should be careful with "synonym replacement". E.g. page 5 line 195 "oncological patients" is one such example. There are numerous such inappropriate synonym replacements. Authors should revise their write up extensively

Thank you for your valuable comment. We revised our manuscript. It has been corrected grammatically and stylistically.

  1. More figures should be included. A figure summarizing the methods for exosome/microparticle/microvesicle, whichever term authors choose finally (see below my comment), detection should be included. Another "timeline figure" should summarize the significant discoveries chronologically.

As recommended by the Reviewer, we have added additional figures. One concerns the path of microparticles from discovery to definitione - Figure 1. The second is a table summarizing the current methods of detecting microparticles - Table 1.

Specific comments

  1. The choice of term "microparticles" is ambiguous and confusing, slightly arbitrary. Authors should choose one term (e.g., micropartciles, or microvesicales), cite adequate references to support their choice and then stick to it throughout text

Thank you for your valuable comment. We have specified our nomenclature to the word Microparticles. Our selection is confirmed by the literature from verse 91-99. This will be quoting 6,7,19. Researchers still have a problem with the nomenclature. In our article, we take into account the difficulties of researchers with the nomenclature, but we focus mainly on the MP.

  1. The paper is not formatted like a review paper. Also section numbering needs to be carefully checked

Thank you for this comment. We have corrected the titles of our paragraphs. We responded to the requirements of the Biomedicines journal. Our article consists of a summary, introduction, and main text, in which we have provided an overview of the research, discussions and conclusions.

  1. Page 3 last pargraph (line 129-142). The information is inaccurate. Not all exosomes need external stimulus for release. Extensive literature is available on this topic and many excellent reviews have been published. Authors should consult published papers (e.g., including but not limited to the excellent review by Hessvik et al. Cell Mol Life Sci. 2018; 75(2): 193–208)

Thank you for your comment. The paragraph that covers lines 117-142 has been updated. The fact is that not all exosomes need an external stimulus to release. In our work, we quoted, among others, Hessvik et al. Cell Mol Life Sci. 2018; 75 (2): 193–208. The information in this article proved to be of great use in updating our knowledge.

  1. Page 4 section 3.4: NTA is now much more commonly used. Authors should include more discussion on NTA based exosome detection. Also, a brief discussion on modern electrochemical and microfluidics based platforms will also be useful

We agree with the Reviewer. Verse 194 starts the section on NTA; we have included the most used detection method. From verse 217-247 a short discussion on modern electrochemical and microfluidics based platforms.

  1. Page 4 line 187: Traditional centrifugation is misleading term, plus this is not a detection method. Ultracentrifugation is needed to isolate exosomes. They are then detected by any of the several methods

Thank you for your valuable comment. Lines 170-181 contain information about differential centrifugation and ultracentrifugation as traditional isolation methods. We also explained two processes, namely isolation and detection. Table 1 lists the detection methods.

  1. Section 3.5 is incomplete. This section should highlight in more detail how exosomes/microparticles play role in neoplastic processes, e.g., the possible role played by exosomal cargo (this information is included in first paragraph of 3.6 but it belongs to 3.5)

This information has been moved to the correct section. Now it starts with line 250. We have expanded the subject matter considerably in line with the valuable comment from the reviewer. We have developed a thread about the transport function of microparticles. From line 274, there is a discussion about the transport function of microparticles. In addition, we added a paragraph on immunomodulation from line 295.

  1. There is a large body of research available on the role of exosomes in prostate cancer (e.g., an excellent recent review Int. J. Mol. Sci.2020, 21(6), 2118). Authors have only cited 7 studies which is not sufficient. This section should be expanded to inlcude all the significant contributions done so far in this area.

This review is useful and contributed a lot to our article. We have reviewed the author's research on microparticles as prostate cancer biomarkers. Our review of author research begins with 344. Once again, thank you for your valuable comments and for contributing a lot to our article review.

Yours faithfully

Authors

Reviewer 2 Report

Authors review the importance and possible clinical use of microparticles detection in the diagnosis and prognostic determination of prostate cancer. The subject is interesting and the review is didactical.  

Author Response

Answer for the reviewer

Thanks for your review.

Thank you for reading our manuscript and valuable comments

Yours faithfully

Authors

Round 2

Reviewer 1 Report

Thank you for submitting the revised manuscript. Although authors have tried their best to address this reviewer's comments, there is a basic flaw in the understanding of the authors regarding the main theme of this article. Moreover, the write up is overall very much ambiguous and it is very hard for an average reader to understand what the authors are actually talking about

The authors have chosen the term "microparticles" and defined it on page 3 line 96 as "ectosomes". Therefore it is understood that the paper discusses the role of ectosomes in prosttate cancer. Exosomes and ectosomes are two different types of extracellular vesicles and are generated by completely different mechanisms and play distinct roles (e.g. see Curr Biol. 2018;28(8):R435-R444) However, throughout the paper, authors have mostly discussed about the exosomes and their role in prostate cancer. From the methods of isolation to detection, and the biological role of so-called MPs in cancer, all (or at least most) of the discussion is about exosomes. However, when it comes to biogenesis, authors have mainly discussed the mechanism of ectosome production. Table 1 and Table 2 cite studies on both exosomes and ectosomes/MPs. Figure 1 mentions the structure of exosomes. It appears the authors do not fully understand the difference between various types of EVs.

Moreover, the write-up is not of acceptable quality. There are numerous ambiguous sentences, poor word choices and other grammatical errors which make it really hard to understand what the authors main point is. For example "Thrombosis and Haemostasis of Vascular Biology developed and published an official definition of a micro-particle, which allows for the precise meaning of the cell line under that name" and "Exposure of the coagulating phospholipids and specific receptors on the MP surface causes them to also act as transmitters of inflammation, thrombosis, and angiogenesis."

I think authors need a thorough rethink on the main theme of their paper and once they have selected one, they would need to rewrite the whole paper sticking to that one main theme. E.g. if they want to review literature on MPs/ectosomes, everything; isolation and detection methods, first discovery and over time progress in the field, role in prostate (or any other) cancer, mechanism of biogenesis, all should be focused on MPs/ectosomes. Authors cannot conveniently switch between exosomes and ectosomes.

Author Response

Answer for the Reviewer

We would like to thank the Reviewer for very valuable comments. Detailed answers to individual items are provided below.

We have revised the following lines:

  1. 72-79 - Specified methods of the first selection of articles
  2. 94-97, 159-160. -Corrected a sentence incomprehensible to the reviewer
  3. 98-121 -New rationale for the nomenclature used in the work
  4. 195-201- Specifying flow nanocytometry instead of flow cytometry

The content of the article was consulted with a laboratory worker.

Yours faithfully

Authors

Round 3

Reviewer 1 Report

Authors have addressed my comments satisfactorily. However, use of professional services to improve language and grammar is recommended.

Author Response

Thank you very much for revising our manuscript.

We verified the English language. The manuscript was sent to a native English-speaking physician who revised it linguistically. The content of the article and the content of the tables have been improved.
